# Ensemble-Learning and Feature Selection Techniques for Enhanced Antisense Oligonucleotide Efficacy Prediction in Exon Skipping

**DOI:** 10.3390/pharmaceutics15071808

**Published:** 2023-06-24

**Authors:** Alex Zhu, Shuntaro Chiba, Yuki Shimizu, Katsuhiko Kunitake, Yasushi Okuno, Yoshitsugu Aoki, Toshifumi Yokota

**Affiliations:** 1Phillips Academy, Andover, MA 01810, USA; azhu23@andover.edu; 2Department of Medical Generics, Faculty of Medicine and Dentistry, University of Alberta, Edmonton, AB T6G 2H7, Canada; 3HPC- and AI-Driven Drug Development Platform Division, RIKEN Center for Computational Science, Yokohama 230-0045, Japan; shuntaro.chiba@riken.jp (S.C.); okuno.yasushi.4c@kyoto-u.ac.jp (Y.O.); 4Department of Biomedical Data Intelligence, Graduate School of Medicine, Kyoto University, Kyoto 606-8507, Japan; shimizu.yuki.75s@st.kyoto-u.ac.jp; 5Department of Molecular Therapy, National Institute of Neuroscience, National Center of Neurology and Psychiatry (NCNP), Kodaira, Tokyo 187-8551, Japan; kunitake-k@ncnp.go.jp (K.K.); tsugu56@ncnp.go.jp (Y.A.)

**Keywords:** antisense oligonucleotides, exon skipping, machine learning, ensemble learning, personalized medicine, n-of-1 therapy, splice switching, genetic disease, splicing, RNA

## Abstract

Antisense oligonucleotide (ASO)-mediated exon skipping has become a valuable tool for investigating gene function and developing gene therapy. Machine-learning-based computational methods, such as eSkip-Finder, have been developed to predict the efficacy of ASOs via exon skipping. However, these methods are computationally demanding, and the accuracy of predictions remains suboptimal. In this study, we propose a new approach to reduce the computational burden and improve the prediction performance by using feature selection within machine-learning algorithms and ensemble-learning techniques. We evaluated our approach using a dataset of experimentally validated exon-skipping events, dividing it into training and testing sets. Our results demonstrate that using a three-way-voting approach with random forest, gradient boosting, and XGBoost can significantly reduce the computation time to under ten seconds while improving prediction performance, as measured by *R*^2^ for both 2′-O-methyl nucleotides (2OMe) and phosphorodiamidate morpholino oligomers (PMOs). Additionally, the feature importance ranking derived from our approach is in good agreement with previously published results. Our findings suggest that our approach has the potential to enhance the accuracy and efficiency of predicting ASO efficacy via exon skipping. It could also facilitate the development of novel therapeutic strategies. This study could contribute to the ongoing efforts to improve ASO design and optimize gene therapy approaches.

## 1. Introduction

Antisense oligonucleotides (ASOs) have emerged as a powerful tool in the field of molecular biology and have attracted widespread attention as a promising therapeutic modality for a range of genetic diseases. These small, single-stranded nucleotides function by binding to the complementary sense strand of specific mRNAs through Watson–Crick base pairing, leading to the modulation of gene expression through a variety of mechanisms [1]. The therapeutic potential of ASOs was recognized in the 1970s [2]. However, early versions of unmodified ASOs were found to have limited plasma persistence and bioavailability, which posed significant challenges to their clinical utility [3]. 

Over the years, ASOs have undergone three generations of development to improve their stability, bioavailability, and binding affinity. These advancements have been achieved through the modification of sugar moieties, bases, and phosphodiester linkages. The first generation of ASOs involved the use of unmodified nucleotides, which were unstable and rapidly degraded in vivo. This led to the development of the second-generation ASOs, which incorporated 2′-O-methyl nucleotides (2OMe) to enhance their stability and binding affinity [4]. The third generation of ASOs is represented by phosphorodiamidate morpholino oligomers (PMOs), which contain a neutral backbone and show improved cellular uptake and bioavailability compared to previous generations [4].

ASOs modify target mRNA expression through two main mechanisms: RNase H-dependent cleavage and steric block [5]. RNase H-dependent ASOs, designed as gapmers, bind to the target RNA and trigger cleavage by the endogenous RNase H enzyme, leading to target gene silencing [6,7,8]. Steric blocking ASOs, on the other hand, are often employed to specifically exclude (exon skipping) or retain (exon inclusion) a specific exon(s), leading to alternations in splicing decisions [2,9].

Phosphorothioates also play a significant role in ASOs and have contributed to the development of ASO-based therapies. Fomivirsen, the first antisense drug approved by the U.S. Food and Drug Administration (FDA), is an excellent example of the application of phosphorothioates in ASOs [10].

The improvements achieved in ASO technology have significantly expanded their therapeutic potential and led to numerous successful clinical trials for the treatment of various genetic disorders. For example, nusinersen, an exon-inclusion 2′-O-Metoxyethyl-modified ASO, was approved by the U.S. FDA in 2016 for the treatment of spinal muscular atrophy (SMA), a devastating neuromuscular disease that is caused by the loss of function of the survival motor neuron 1 (SMN1) gene [11]. Similarly, eteplirsen, an exon-skipping PMO ASO, was approved in 2016 for the treatment of Duchenne muscular dystrophy (DMD), a lethal X-linked disorder that leads to progressive muscle wasting and early mortality [12].

Exon skipping, where an ASO causes the exclusion of a specific exon in splicing, has emerged as a promising treatment for genetic diseases, especially muscular dystrophies. The U.S. FDA has approved multiple exon-skipping ASO treatments for DMD, including eteplirsen, golodirsen, viltolarsen, and casimersen [13,14,15,16]. These ASOs induce exon skipping, which leads to the production of a truncated but still functional dystrophin protein. For example, eteplirsen targets exon 51, while viltolarsen and golodirsen target exon 53, and casimersen targets exon 45. By inducing exon skipping, these ASOs enable the production of a shortened but functional dystrophin protein, which can partially restore muscle function and slow disease progression.

Exon skipping has shown promising potential as a treatment option for many genetic diseases beyond DMD. Splicing defects are a common cause of many genetic diseases, and exon skipping can be used to restore proper splicing by skipping over faulty exons. Milasen, a patient-customized n-of-1 ASO drug targeted for a pseudoexon in the neuronal ceroid lipofuscinosis-7 (CLN7) gene, was recently approved by the FDA for the treatment of a single patient with Batten’s disease, demonstrating the potential of exon skipping for personalized medicine [17,18]. Milasen is an ASO that targets a pseudoexon with a novel intronic mutation in the CLN7 gene, which encodes a protein involved in lysosomal function [19]. Milasen targets a complementary RNA sequence in the pseudoexon, leading to the production of a full-length CLN7 protein. This approach is an example of personalized medicine, where the ASO is tailored to the specific genetic mutation present in the patient. Exon-skipping therapies are also being explored for other genetic diseases, such as cystic fibrosis [20], retinitis pigmentosa [21], sarcoglycanopathy [22,23], dysferlinopathy [24,25,26], fibrodysplasia ossificans progressive [27,28], epidermolysis bullosa [29,30], frontotemporal dementia with, parkinsonism linked to chromosome 17 (FTDP-17) [31,32], and cancer [33], among others.

Despite these promising developments, there are still significant challenges in developing effective exon-skipping therapies. A major hurdle is the difficulty in selecting an optimal sequence for exon skipping, as the efficacy of ASOs is often unpredictable due to numerous factors involved in the exon-skipping process [34]. Designing effective ASO sequences requires consideration of various criteria [35], particularly for exon skipping [36]. Software tools, such as eSkip-Finder, can aid in this process [37]. eSkip-Finder (https://eskip-finder.org, accessed on 1 May 2023) is a web-based tool developed by Chiba et al. that provides a solution for identifying optimal ASO sequences for exon skipping by using machine-learning models built from a curated database of publications and patents [37]. 

The selection of important features is a crucial step in the tool’s approach, and the eSkip-Finder uses an exhaustive search of subsets of features to identify these critical components. However, due to the high computational cost, the subset size was limited to seven features. To optimize the performance of the models, hyperparameters in the support vector regressor are optimized through a grid search. This optimization process is computationally intensive, requiring a significant amount of computing power, and can take several days to complete.

This paper seeks an alternative solution to reduce the computational cost associated with the eSkip-Finder. Some machine-learning algorithms, such as decision tree or random forest, have built-in feature-ranking capabilities [38]. Ensemble methods are also proven to have good performance with reasonable computation cost [39,40]. We explored their utility in ASO efficacy prediction and demonstrated that a combination of three algorithms, namely random forest, gradient boosting, and XGBoost, through a three-way voting mechanism, can significantly reduce computation time while maintaining or slightly improving the prediction performance. This approach offers a promising solution for reducing computational cost in the ASO efficacy prediction process.

## 2. Materials and Methods

### 2.1. Dataset Description

The datasets utilized in this study were identical to those employed in Chiba et al. [37]. For PMO, 369 and 57 measurements were used for training and testing, respectively, and there were 98 and 11 unique ASO sequences in each split without any overlap. Similarly, for 2OMe, 197 and 31 measurements were used for training and testing, respectively, with 111 and 13 unique ASO sequences in each split without overlapping. Given that PMO and 2OMe exhibit different chemical properties and binding affinities, the datasets were treated separately throughout the analysis. 

### 2.2. Feature Description 

For each measurement, there were 32 numerical features calculated via bioinformatics tools, as discussed in Chiba et al. (such as dose). The categorical feature, Malueka’s category, was excluded from modeling. As reported in [37], this feature was not important in determining the ASO efficacy and was specifically linked to dystrophin exons [41]. Models developed with this feature included would be difficult to generalize to other genes.

### 2.3. Problem Formulation and Model Input

The efficacy was measured as a percent in the range of 0 to 100, both inclusive. We wanted to develop a machine-learning model to predict the efficacy value of a given ASO with associated feature vector, which makes it a regression problem. All 32 features were inputted into the machine-learning models, and feature selection was left to the models themselves.

### 2.4. Machine-Learning Libraries and Regressors

The machine-learning libraries included scikit-learn (0.42.2) [42] and XGBoost (1.6.1) [43]. The following regressors were used: support vector, random forest, gradient boosting, and XGBoost. The last three were also used to vote by the simple average of the individual predictions. The support vector regressor was included for comparison purposes, as it was used in Chiba et al. All the regressors were built without hyperparameter tuning, i.e., default parameters were used in each regressor (except random seeds). The computation code was developed using Python (3.9.7) on Mac (Quadcore i5, 2 GHz CPU, 16 GB RAM).

### 2.5. Model Assessment and Selection

Two metrics were used to assess model performances: *R*^2^ and mean absolute error (MAE) between true efficacy values and predictions. The models were first assessed on the training data via 10-fold cross-validation. Other numbers of folds were also attempted, but they gave similar results. The best model was then selected and applied to the reserved test data. The *R*^2^ and MAE on each fold were collected, and their mean and standard deviation were further computed to aid the best model selection. The model with the highest *R*^2^ and lowest MAE values was considered the best-performing model. 

### 2.6. Feature Importance Analysis

While the random forest, gradient boosting, and XGBoost models were trained, they also collected data to compute the feature importance score. The voting regressor had no feature importance score; however, we used the model-agnostic method, permutation feature importance provided by scikit-learn, to rank the feature importance. This analysis helped identify the most significant features contributing to efficacy prediction and provided insights into the underlying biological processes related to ASO efficacy. 

### 2.7. Model Comparison and Generalizability

To further assess the performance of the proposed ensemble approach and its individual components (random forest, gradient boosting, and XGBoost), we compared the results with the support vector regressor, as utilized in Chiba et al. This comparison aimed to validate the effectiveness of the ensemble method in terms of prediction accuracy, computational efficiency, and generalizability. 

To further access the potential generalizability of the predictive models, we applied the PMO model to a gene not seen in the training dataset (the exon 73 skipping of collagen type VII alpha 1 chain). We compared the efficacy ranking order from prediction to the real experimental measurements. 

## 3. Results

The performance metrics for various models using 10-fold cross-validation on the training data are shown in Table 1. The five-fold and twenty-fold cross-validations were also attempted, and the results were similar to what was reported here. The data splitting was based on ASOs, i.e., there were no overlapping ASOs in training and validation splits. As can be seen from Table 1, for both PMO and 2OMe ASOs, the three-way-voting approach gives the largest *R*^2^ and smallest MAE. We thus chose this approach and applied it to the test datasets. The support vector regressor performed noticeably poorly as there was no hyperparameter optimization in the current study. It shall also be noted that the whole computing took about 10 s on a laptop computer.

When the three-way-voting models, trained on the training data with all features, were applied to the test data, the predictions were similarly assessed. For PMO, we have *R*^2^ = 0.706 and MAE = 12.250 and for 2OMe, *R*^2^ = 0.795 and MAE = 9.237. The *R*^2^ values are higher than those reported [37], which were 0.6 and 0.7, respectively. The true efficacy and predicted one have a good linear correlation, as depicted in Figure 1. It shall be noted that unlike the support vector regressor, which can generate unrealistic, negative efficacy values, the three-way voting approach will not possibly predict a negative efficacy as long as the input data has no negative efficacy.

The feature importance ranking using the training data as reported by the three-way voting is shown in Figure 2. The rankings using the test data are similar on top-ranked features, suggesting that overfitting is not a concern. Among the top five and ten features using training or test dataset, three (ACP, oligo concentration, dG (100BaseFlanks, RNAstructure)) and eight (ACC_AVE, ACC_LAST8, ACP, distance from acceptor (position of last base relative to acceptor), length, oligo concentration, dG (100BaseFlanks, RNAstructure), dG (200BaseFlanks, RNAstructure)) are common for PMO, and four (# exon GCs blocked by oligo, %GC of exon when blocked by oligo, ACP, Oligo concentration) and nine (# exon GCs blocked by oligo, %GC of exon when blocked by oligo, ACC_LAST15, ACP, distance from donor (position of first base relative to donor), oligo concentration, dG (50BaseFlanksAroundTarget, RNA structure), dG (TargetAsExon, RNAstructure), niscore) are common for 2OMe. The four PMO features (oligo concentration, exon v intron %GC after blocking by oligo, dG (50BaseFlanksAroundTarget), ACC LAST15) used in Chiba et al. here were ranked at 1, 24, 11, and 15. The 6 2OMe features (oligo concentration, GCs (number of), ACP, %GC of exon when blocked by oligo, niscore per base, ACC LAST8) used in Chiba et al. here were ranked at 2, 25, 4, 3, 17, and 11. In both cases, some correlation can be observed. We also noted that some features were strongly correlated, as shown in Figure 3. As an example, niscore and niscore_per_base are strongly correlated. Niscore_per_base was ranked seventeenth, but niscore was ranked fifth in our 2OMe model. Therefore, at least some discrepancies can be attributed to the feature correlations. Due to the randomness in the algorithms, the rank order can be slightly different in each run. We also did not filter out strongly correlated features as the cut-off threshold for correlation coefficient is to some extent arbitrary.

With the above feature importance ranking, we used top k (k = 1, 2, …, 32) features to do the 10-fold cross validation with three-way voting, similar to the experiment that generates the data in Table 1, except top k features were used instead of all 32 features. The results are shown in Figure 4. As can be seen, for PMO, top 8–15 features give the best *R*^2^ and for 2OMe, top six and more features give the best results. The variation, specifically in the PMO case, can be attributed to the randomness in data split. Using the top features sometimes improves the predictive performance on the test dataset. Since the behavior is not consistent for both PMO and 2OMe and it is also difficult to pick a reasonable k for PMO, we decided not to explore it further to reduce the risk of the test data leaking into the model development. 

To check if the voting approach works for different genes and exons, we applied the trained PMO model to the exon 73 skipping of collagen type VII alpha 1 chain [9]. The results are summarized in Table 2. The predictions by the voting approach preserve the ranking order of ASO efficacy experimentally measured. Cautions must be taken when one extends the model to a different application domain however. As more data is accumulated in databases, such as eSkip-Finder, we expect predictive models will be validated rigorously and extended as needed.

## 4. Discussions

In this study, we applied machine-learning algorithms with built-in feature selection capabilities to train and predict the exon-skipping efficacy of PMO and 2OMe ASOs. The results of this study indicate that the three-way-voting ensemble approach using random forest, gradient boosting, and XGBoost regressors outperforms the support vector regressor in terms of prediction accuracy for both PMO and 2OMe ASOs. The improved performance is evident through higher *R*^2^ values and lower MAE in both training and test datasets. For PMO, we have *R*^2^ = 0.706 and MAE = 12.250, and for 2OMe, *R*^2^ = 0.795 and MAE = 9.237. The *R*^2^ values are higher than those in the current eSkip-Finder model, which were 0.6 and 0.7, respectively [37]. The support vector regressor performed poorly in this study, likely due to the lack of hyperparameter optimization. Additionally, the ensemble approach was computationally efficient, requiring only 10 s for computation on a laptop computer. 

The ensemble approach presented in this study offers several advantages over the support vector regressor, including improved prediction accuracy, computational efficiency, and generalizability. The improved performance and versatility of this model make it a valuable tool for designing novel ASOs for exon skipping, optimizing existing ASO therapies, and developing personalized medicine approaches. The true efficacy and predicted efficacy values demonstrated a strong linear correlation, and the three-way-voting approach did not predict any negative efficacy values. The feature importance rankings were consistent across training and test datasets, suggesting minimal overfitting. Although some discrepancies in feature rankings were observed compared to Chiba et al., these differences can be attributed to feature correlations and the randomness inherent in the algorithms. 

When the PMO model was applied to a different gene, exon 73 skipping of collagen type VII alpha 1 chain, the three-way voting approach was able to preserve the ranking order of ASO efficacy. However, caution should be exercised when extending the model to different application domains as the model’s performance may be influenced by differences in target genes or exons.

The study emphasizes the importance of feature selection in developing accurate predictive models for ASO efficacy. Feature selection is a critical step in machine learning as it helps to identify the most informative and relevant features for predicting the target variable. We used three different methods, each identifying the most important features for predicting exon-skipping efficacy of PMO and 2OMe ASOs. The feature importance ranking generated by the three-way-voting approach revealed the top features used in the prediction of exon-skipping efficacy for both PMO and 2OMe ASOs. These findings suggest that the selection of informative features is crucial for developing accurate and interpretable predictive models for ASO efficacy.

The study highlights the potential applications of the developed predictive models for drug development and personalized medicine. ASOs have emerged as a promising therapeutic strategy for a wide range of diseases, including DMD, Batten’s disease, and retinitis pigmentosa [44,45,46]. The ability to predict ASO efficacy accurately and efficiently could accelerate the drug development process by enabling researchers to identify the most promising ASOs for further development. Moreover, personalized medicine approaches could be developed by using predictive models to select ASOs that are most likely to be effective for specific patients based on their genetic profiles.

The study provides insights into the limitations and challenges of the developed predictive models. One potential limitation of the voting approach is that it relies on engineered features hand-picked by scientists. Although most selected features were found to be consistent with previous studies and eSkip-Finder, there is still a possibility that important features have been overlooked or excluded. Moreover, the voting approach may not generalize well to other diseases or target regions, and further validation is required to ensure the applicability of the approach. Additionally, the study focused on predicting exon-skipping efficacy of PMO and 2OMe ASOs, and the performance of the developed models for other types of ASOs needs to be evaluated in future studies. As a possible future extension, one could consider machine-learning algorithms in combination with natural language-processing techniques, which has been successfully applied to biological sequence analysis [47].

As mentioned above, the voting approach predicts non-negative efficacies as long as there are no samples with negative efficacies in the training data. This aspect of the voting approach warrants further discussion as it has important implications for the interpretation of the predicted efficacies. By design, the voting approach ensures that no negative efficacies are predicted, which is a desirable property since negative efficacies are not biologically meaningful. However, this also means that the approach will not predict any efficacies larger than the highest efficacy in the training data since decision trees are used essentially in the individual algorithms. However, this can be a drawback, i.e., the approach will not predict any efficacies larger than the highest efficacy in the training data since decision trees are used essentially in the individual algorithms. While the approach has demonstrated promising results in predicting exon-skipping efficacy of PMO and 2OMe ASOs, its performance is constrained by the training data and may not be able to predict efficacies that are outside the range of the training data. Further research is needed to validate the approach and to compare its performance with other machine-learning algorithms.

The proposed voting approach has a very short training time. The short training time of the voting approach is a significant advantage of the method as it enables rapid development of predictive models for ASO efficacy. In the study, we reported that the whole computing took about 10 s on a laptop computer, which is a remarkable achievement considering the complexity of the problem and the large number of features involved. The short training time of the voting approach is particularly advantageous for drug development, where time and resources are often limited. The ability to rapidly develop predictive models for ASO efficacy could accelerate the drug development process by enabling researchers to identify the most promising ASOs for further development. Moreover, the short training time could also facilitate the development of personalized medicine approaches by enabling rapid screening of ASOs for specific patients based on their genetic profiles.

Future research directions include incorporating additional features, integrating advanced machine-learning techniques, such as natural language-processing techniques, as mentioned above, and applying the model to different types of ASOs and diseases. As more data become available in databases, such as eSkip-Finder, predictive models can be validated more rigorously and extended as needed, further improving the accuracy and applicability of ASO efficacy predictions. Many machine-learning and artificial intelligence techniques can be applied to drug discovery. For a recent review, please refer to [48].

In conclusion, the study presents a promising approach for predicting exon-skipping efficacy of PMO and 2OMe ASOs using machine-learning algorithms with built-in feature selection capabilities. The findings emphasize the importance of feature selection and have potential applications for drug development and personalized medicine. However, further validation is required to ensure the applicability of the approach for other diseases and ASO types. The study also highlights the potential for integrating machine-learning algorithms with natural language-processing techniques for biological sequence analysis, which could provide a more comprehensive understanding of ASO-mediated exon skipping.

## Figures and Tables

**Figure 1 pharmaceutics-15-01808-f001:**
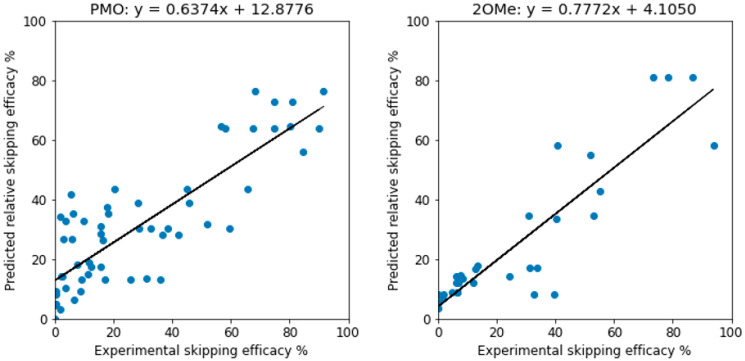
Predictive performance of three-way voting for PMO (left) and 2OMe (right) ASOs. When the three-way-voting approach was applied to the test data, we observed improved predictive performance for both PMO and 2OMe AOs compared to previous studies.

**Figure 2 pharmaceutics-15-01808-f002:**
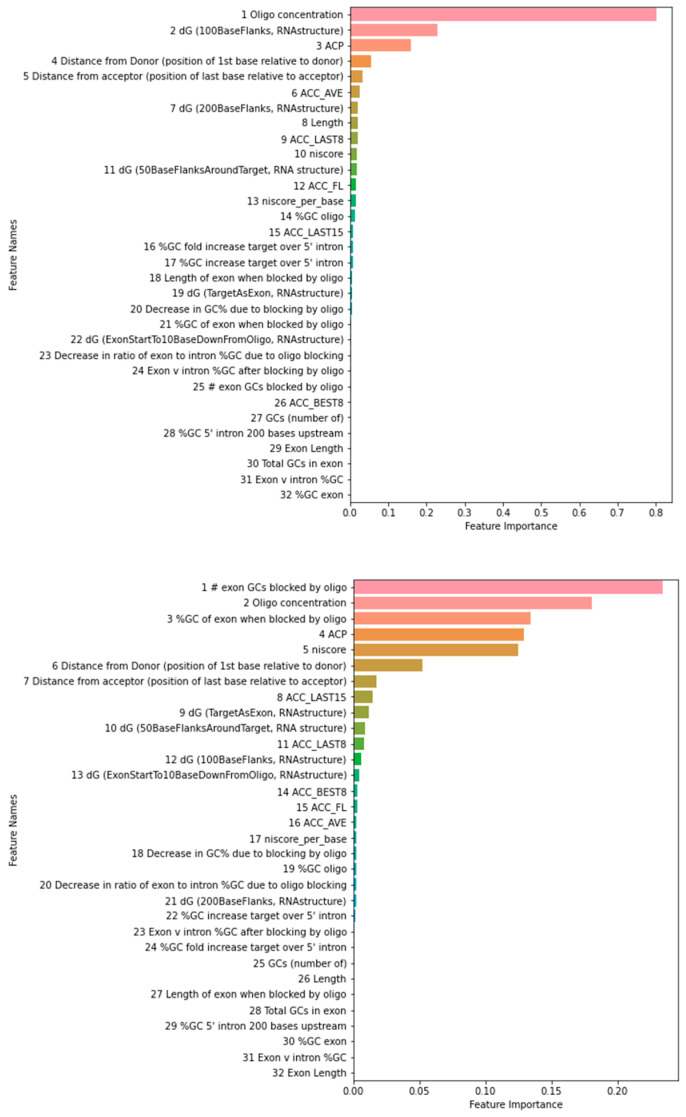
Feature importance as determined by the three-way-voting method. The feature importance scores for PMO and 2OMe are displayed on the top and bottom sides of the figure, respectively. Higher scores indicate greater importance of the feature for predicting exon-skipping efficacy. #; number.

**Figure 3 pharmaceutics-15-01808-f003:**
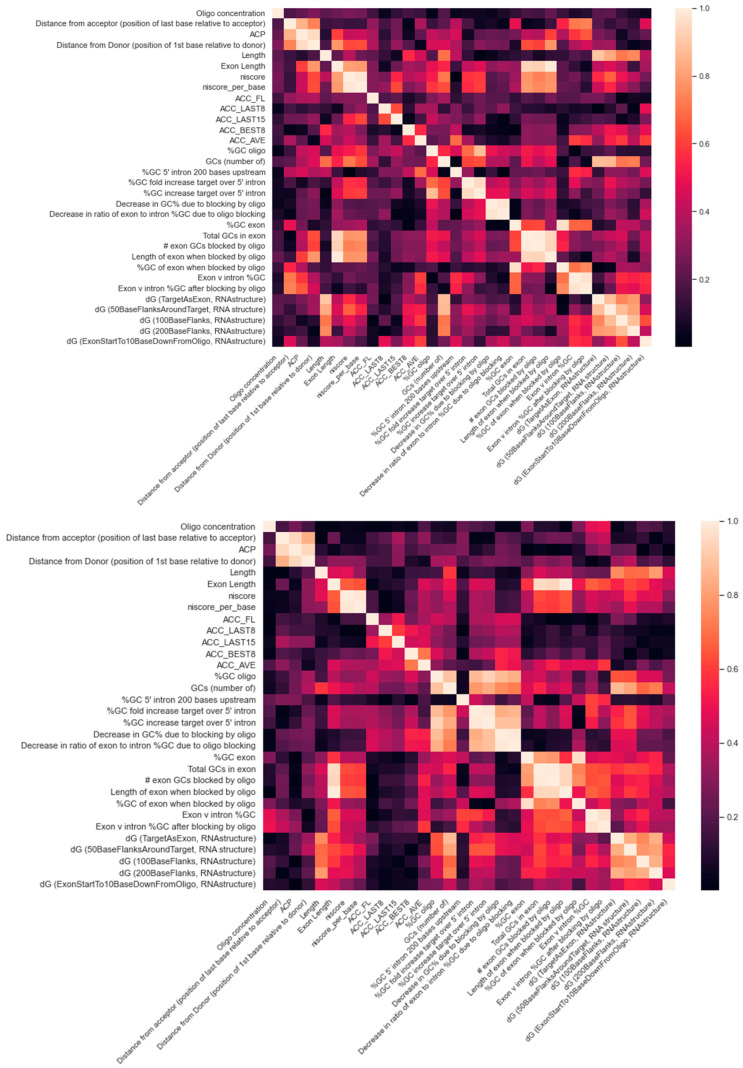
Feature correlations. The feature correlations (absolute values) for PMO and 2OMe are displayed on the top and bottom sides of the figure, respectively. #; number.

**Figure 4 pharmaceutics-15-01808-f004:**
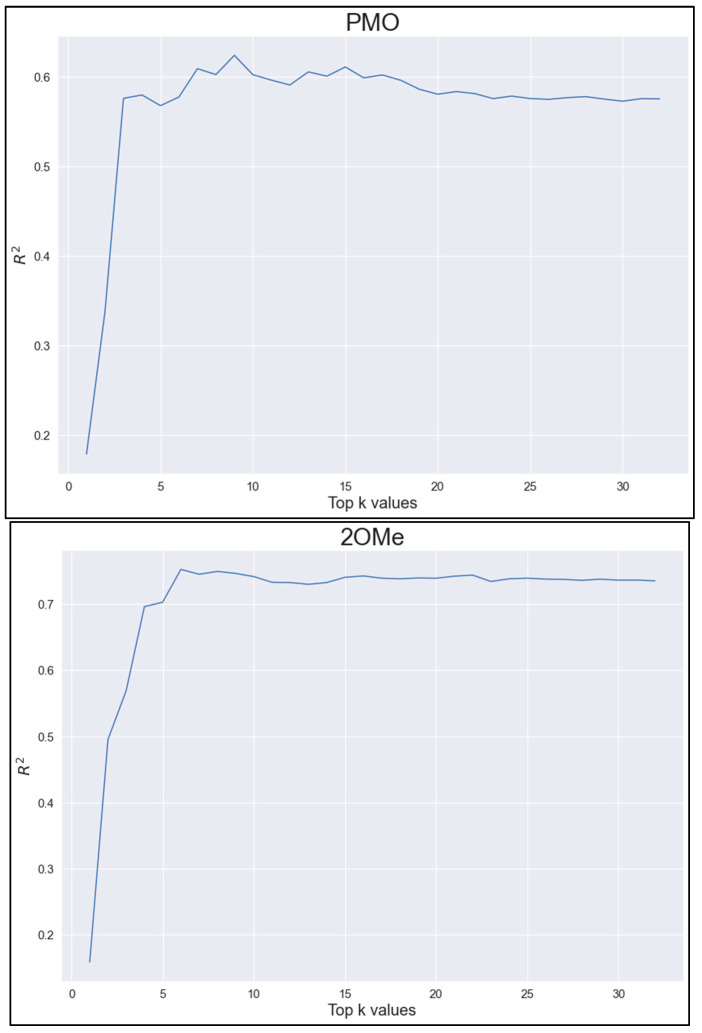
Top k features. *R*^2^ as a function of top k features when top k features were used for 10-fold cross validation on the training dataset (top for PMO and bottom for 2OMe).

**Table 1 pharmaceutics-15-01808-t001:** Model performance assessed on training datasets with 10-fold cross-validation.

Methods	PMO		2OMe	
	*R* ^2^	MAE	*R* ^2^	MAE
Support Vector	0.138 ± 0.076	22.06 ± 4.02	0.558 ± 0.093	17.70 ± 5.32
Random Forest	0.555 ± 0.247	15.39 ± 4.84	0.729 ± 0.169	10.59 ± 3.31
Gradient Boosting	0.564 ± 0.234	14.97 ± 4.58	0.721 ± 0.152	10.13 ± 2.77
XGBoost	0.530 ± 0.214	15.58 ± 3.87	0.717 ± 0.164	10.56 ± 3.49
Three-way Voting	0.576 ± 0.244	14.87 ± 4.63	0.740 ± 0.157	10.07 ± 3.29

The uncertainty represents standard deviation of 10-fold cross validation.

**Table 2 pharmaceutics-15-01808-t002:** Prediction of exon 73 skipping of collagen type VII alpha 1 chain using PMOs.

ASO Name	Voting Predicted	eSkip Predicted	Experimental [14]
H73A (+16 + 40)	63% (ranked #1)	60% (ranked #1)	100% (ranked #1)
H73A (+16 + 35)	37% (ranked #3)	23% (ranked #3)	40% (ranked #3)
H73A (+21 + 40)	42% (ranked #2)	48% (ranked #2)	85% (ranked #2)

## Data Availability

The data used in this study can be accessed from [49]. No new data were created.

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
