# Peer review of "Ensemble-Learning and Feature Selection Techniques for Enhanced Antisense Oligonucleotide Efficacy Prediction in Exon Skipping"

_pharmaceutics, 2023, doi:10.3390/pharmaceutics15071808_

Round 1

Reviewer 1 Report

In this manuscript, Zhu et al applied machine learning algorithms such as random forest, gradient boosting, and XGBoost with built-in feature selection capabilities to train and predict the exon-skipping efficacy of phosphorodiamidate morpholino oligomers and 2’-O-methyl nucleotides antisense oligonucleotides

The topic of the submission is timely, with a tremendous interest being placed currently in drug development and personalized medicine. The authors using eSkip-Finder and ensemble approaches tried to minimize computational burden and develop accurate predictive models for ASO efficacy.

As a general comment the authors used one dataset of experimentally 22 validated exon skipping events to evaluate their approach. Is the data sufficient for a ML model and to draw conclusions?

Specific comments

Line 92, the authors gave numerous references stressing exon skipping therapies however just putting in parenthesis references [19-32] is not acceptable for the introduction section. I suggest you improve this point by discussing these references or minimize them but present this information in detail.

The quality of Figures 3 and 4 is not good. It must be improved

Perhaps, a short paragraph summarizing the main limitations of this approach or ML similar techniques, would further improve the scientific value of the paper.

The manuscript should also highlight further artificial intelligence and machine learning methods in the literature. I believe it is necessary to categorically examine the studies on these and discuss the results. The datasets used in these studies could be included in this discussion.

Conclusions are consistent with the evidence presented. Perhaps the authors could present some specific targets for future studies.

Minor editing of English language required, for example on page 5 (results) the paragraph needs better reorganization.

Author Response

Reply to Reviewer 1

Thank you for the useful and constructive comments. We considered those carefully when revising the manuscript. Following are our reply to your comments/suggestions.

Comment:  As a general comment the authors used one dataset of experimentally 22 validated exon skipping events to evaluate their approach. Is the data sufficient for a ML model and to draw

conclusions? ?

This is a very good question and every machine learning work should carefully consider it. First of all, as mentioned in our manuscript, 369 and 57 measurements of exon skipping were used for training and testing for PMOs, respectively, and for 2OMe, 197 and 31 measurements were used for training and testing, respectively. When training models, one should pay attention to potential overfitting and when applying models, one should be careful with generalization. We looked at the possible overfitting issue and concluded it is not a concern (The first paragraph after Figure 1: “The rankings using the test data are similar on top-ranked features, suggesting that overfitting is not a concern”. In the Discussion, “The feature importance rankings were consistent across training and test datasets, suggesting minimal overfitting”. ). The experiment described at the end of the Results is just designed to check the generalizability: “To check if the voting approach works for different genes and exons, we applied the trained PMO model to the exon 73 skipping of collagen type VII alpha 1 chain” – the preliminary results suggest a good generalizability. However, we do remind the readers, in the same paragraph, “Cautions must be taken when one extends the model to a different application domain, however. As more data is accumulated in databases such as eSkip-Finder, we expect predictive models will be validated rigorously and extended as needed”.

 comment:

Line 92, the authors gave numerous references stressing exon skipping therapies however just putting in parenthesis references [19-32] is not acceptable for the introduction section. I suggest you improve this point by discussing these references or minimize them but present this information in detail.

 We have revised this section and cited references individually for each disease.

Comment: The quality of Figures 3 and 4 is not good. It must be improved

We have improved the quality (larger font size) and are submitting original figures.

Comment: Perhaps, a short paragraph summarizing the main limitations of this approach or ML similar techniques, would further improve the scientific value of the paper. The manuscript should also highlight further artificial intelligence and machine learning methods in the literature. I believe it is necessary to categorically examine the studies on these and discuss the results. The datasets used in these studies could be included in this discussion.

The limitations of our approach are discussed in the Discussion, associated with feature engineering (“One potential limitation of the voting approach is that it relies on engineered features hand-picked by scientists”) and voting (“the approach will not predict any efficacies larger than the highest efficacy in the training data”). We also discussed potential improvements. For example, “one could consider machine learning algorithms in combination with natural language processing techniques, which has been successfully applied to biological sequence analysis”. We do have interest in researching further along the line. Many AI techniques exist and have been applied to medicine. It is hard to synthesize and discuss the results in our research context. Instead, we added a reference [53 Chen, W.; Liu, X.; Zhang, S.; Chen, S. Artificial intelligence for drug discovery: Resources, methods, and applications. Molecular Therapy: Nucleic Acids 2023, 31, 691-702.].

comment:  Conclusions are consistent with the evidence presented. Perhaps the authors could present some specific targets for future studies.

We have listed a few potential directions to explore further in the Discussion.

Reviewer 2 Report

The article entitled: Improved Prediction of Antisense Oligonucleotide Efficacy for Exon Skipping Using Ensemble Learning and Feature Selection has taken into consideration the important problem from the point of antisense strategy. The ASO is one of the most promising even though the story of it begins in the 70s’ of last century. I appreciated that the authors have paid attention to  2’-O-methyl nucleotide analogs. However in my opinion the significance of phosphorothioates in ASO should be discussed. Fomivirsen is the first antisens drug. Moreover, the extended description of the “experimental part” is highly demanded. In its current form, it is lapidary. Authors also use the term “drug” which is inappropriate in the machine learning context it is better to use term: potentially active molecule etc. The graphs are unreadable (please use higher resolution and simplification). In conclusion, I saw the potential of this manuscript but in its current form, I can not recommend it for publication. Also due to the small number of “trials” I recommend indicating in the title that it is just only the preliminary study. 

Author Response

Reply to Reviewer 2

Thank you for the useful and constructive comments. We considered those carefully when revising the manuscript. Following are our replies to your comments/suggestions.

Comment: I appreciated that the authors have paid attention to 2’-O-methyl nucleotide analogs. However in my opinion the significance of phosphorothioates in ASO should be

discussed. Fomivirsen is the first antisens drug.

We have added a paragraph as suggested.

 comment: Moreover, the extended description of the “experimental part” is highly demanded. In its current form, it is lapidary.

We have updated and added more details to Section 2, particularly Section 2.3, 2.5, and 2.7.

 comment: Authors also use the term “drug” which is inappropriate in the machine learning

context it is better to use term: potentially active molecule etc.

We appreciate your clear distinction between drugs and “potentially active molecules”. We reviewed our use of the term and made the correction as needed. Most of the remaining use is related to “drug development”, which is a standard term in the pharmaceutical industry.

comment: The graphs are unreadable (please use higher resolution and simplification).

We updated the figures and used larger font sizes. The original figures are also included instead of embedded in the text.

comment: In conclusion, I saw the potential of this manuscript but in its current form, I can not recommend it for publication. Also due to the small number of “trials” I recommend indicating in the title that it is just only the preliminary study.

We appreciate the sensitivity on the dataset size and the generalizability of the model. When training models, one should pay attention to potential overfitting; when applying models, one should be wary of generalization. We looked at the possible overfitting issue and concluded it is not a concern (The first paragraph after Figure 1: “The rankings using the test data are similar on top-ranked features, suggesting that overfitting is not a concern”. In the Discussion, “The feature importance rankings were consistent across training and test datasets, suggesting minimal overfitting”. ). The experiment described at the end of the Results is just designed to check the generalizability: “To check if the voting approach works for different genes and exons, we applied the trained PMO model to the exon 73 skipping of collagen type VII alpha 1 chain” – the preliminary results suggest a good generalizability. However, we do remind the readers, in the same paragraph, “Cautions must be taken when one extends the model to a different application domain, however. As more data is accumulated in databases such as eSkip-Finder, we expect predictive models will be validated rigorously and extended as needed”.
We have also revised the title to address the reviewer’s concern.

Round 2

Reviewer 2 Report

The authors have provided the correct answers to my questions, therefore I can recommend the article for publication.